# Long-Term Evaluation of Dip-Coated PCL-Blend-PEG Coatings in Simulated Conditions

**DOI:** 10.3390/polym12030717

**Published:** 2020-03-24

**Authors:** Anita Ioana Visan, Gianina Popescu-Pelin, Oana Gherasim, Andreea Mihailescu, Marcela Socol, Irina Zgura, Mari Chiritoiu, Livia Elena Sima, Felicia Antohe, Luminita Ivan, Diana M. Vranceanu, Cosmin M. Cotruț, Rodica Cristescu, Gabriel Socol

**Affiliations:** 1Lasers Department, National Institute for Lasers, Plasma and Radiation Physics, Magurele, 077190 Ilfov, Romania; gianina.popescu@inflpr.ro (G.P.-P.); oana.gherasim@inflpr.ro (O.G.); andreea.mihailescu@inflpr.ro (A.M.); rodica.cristescu@inflpr.ro (R.C.); 2Department of Science and Engineering of Oxide Materials and Nanomaterials, Faculty of Applied Chemistry and Materials Science, University Politehnica of Bucharest, 060274 Bucharest, Romania; 3Laboratory of Optical Processes in Nanostructured Materials, National Institute of Materials Physics, Magurele, 077190 Ilfov, Romania; marcela.socol@infim.ro (M.S.); irina.zgura@infim.ro (I.Z.); 4Department of Molecular Cell Biology, Institute of Biochemistry of the Romanian Academy, 060274 Bucharest, Romania; mari.chiritoiu@biochim.ro (M.C.); lsima@biochim.ro (L.E.S.); 5Proteomics Department, Institute of Cellular Biology and Pathology “N. Simionescu” Romanian Academy, 060274 Bucharest, Romania; felicia.antohe@icbp.ro (F.A.); luminita.radulescu@icbp.ro (L.I.); 6Department of Metallic Materials Science, Physical Metallurgy, Faculty of Materials Science and Engineering, University Politehnica of Bucharest, 060274 Bucharest, Romania; diana.vranceanu@upb.ro (D.M.V.); cosmin.cotrut@upb.ro (C.M.C.)

**Keywords:** degradation, electrochemistry, dip-coating, PCL-PEG blends, mass loss

## Abstract

Our study focused on the long-term degradation under simulated conditions of coatings based on different compositions of polycaprolactone-polyethylene glycol blends (PCL-blend-PEG), fabricated for titanium implants by a dip-coating technique. The degradation behavior of polymeric coatings was evaluated by polymer mass loss measurements of the PCL-blend-PEG during immersion in SBF up to 16 weeks and correlated with those yielded from electrochemical experiments. The results are thoroughly supported by extensive compositional and surface analyses (FTIR, GIXRD, SEM, and wettability investigations). We found that the degradation behavior of PCL-blend-PEG coatings is governed by the properties of the main polymer constituents: the PEG solubilizes fast, immediately after the immersion, while the PCL degrades slowly over the whole period of time. Furthermore, the results evidence that the alteration of blend coatings is strongly enhanced by the increase in PEG content. The biological assessment unveiled the beneficial influence of PCL-blend-PEG coatings for the adhesion and spreading of both human-derived mesenchymal stem cells and endothelial cells.

## 1. Introduction

Bioresorbable and biodegradable polymers have been extensively used for the preparation of targeted and controlled drug release systems due to their adjustable specific properties according to the requirements of the concerned cells or tissues [1]. The utilization of polymers in drug delivery formulations provides numerous benefits (e.g., convenient, safe and easy administration, low-cost manufacturing process) [2] that can surpass the pharmacokinetic limitations of conventional dosage forms (e.g., frequent administration in the case of short half-life drugs, difficulty in obtaining a stability state, fluctuations in drug concentration) [3]. In particular, bioresorbable polymeric blends open new perspectives in the fabrication of innovative coatings for implantable devices, as an efficient strategy to upgrade the biofunctionality and performances of conventional medical devices. Therefore, a proper selection of the features related to mechanical strength, tunable and controllable hydrophilicity, and degradability [4] of the polymer coatings will strongly impact on the improvement of metallic implants.

In order to fabricate coatings with controlled degradability properties, we herein propose a method that is not only cost-effective but also suitable for scaling-up the coatings production [5]. The dip-coating (DC) method accounts for a large coverage area and good maneuverability, thus it can be easily deployed in the fabrication of coated implants, irrespective of their geometric shape [6,7].

On the other hand, several literature reports emphasize the crucial impact of polymers’ degradation kinetics on the long-term performance of implants, stressing how it may affect a wide range of processes, from host response, to cell growth and tissue regeneration [8]. An emerging hypothesis in hard tissue restorative and regenerative applications states that an ideal candidate for dental or bone implant must be highly biocompatible and biodegradable, with favorable mechanical properties and a preferential controlled release rate of bioactive molecules [9]. To a large extent, the herein proposed metallic implants modified with coatings based on polycaprolactone-polyethylene glycol polymeric blends (further denoted as: PCL-blend-PEG) fulfill such particular requirements.

Owing to its high permeability for small drug molecules [10], the aliphatic polyester, PCL represents an important candidate for fabricating carriers for drug delivery applications, including: microparticles [11], nanoparticles [12], microspheres [13], coatings [14], and scaffolds [15]. However, due to its intrinsic high hydrophobicity and crystallinity, PCL degrades rather slowly. This restricts its further clinical applications to long term delivery approaches only [16,17,18]. In order to achieve a desired release profile, significant efforts were directed towards exploring PCL’s ability to form compatible blends with other biopolymers, as a way of optimally altering its degradation kinetics. Therefore, in this study, we propose a combination of PCL and the soluble polyethylene glycol (PEG), a polymer reputed for its biocompatibility, tissue reaction, and biodegradation kinetics [19], and, respectively, for its hydrophilicity, nontoxicity, nonimmunogenicity and absence of antigenicity [9,20,21]. The PEG-based materials designed for biomedical applications include scaffolds [22], micelles [23], nanoparticles [24], and coatings [25].

Only a limited number of studies have focused on designing and evaluating the drug-associated PCL/PEG biomaterials and their corresponding degradation, in form of micelles [26], electrospun fibers [27], polymer vesicles [28], scaffolds [29], and nanoparticles [20]. Also, it should be emphasized that little information has been published with respect to the degradation behavior of PCL-based films [30,31]. The reason is related to the difficulty in thoroughly observing the degradation behavior of such low weight biomaterials, as well as to the long-term degradation of coatings.

Therefore, our research contributes to a better understanding of the behavior of PCL-blend-PEG coatings and of their long term stability in simulated body fluid (SBF), when exposed to particular electro-corrosion mechanisms. The biocompatibility of the blend coatings obtained by DC method with different PCL to PEG ratios was assessed on human mesenchymal stem cells (MSC) and endothelial cells (EC), in terms of viability and adhesion. Endothelial cells, which underline the inner surface of the vasculature, play a key role in angiogenesis, and therefore, their interaction with titanium surfaces is an important factor influencing tissue healing. Both MSCs and ECs are known to have an important function during bone healing and regeneration, as they are responsible for injured tissue vascularization and new bone formation. While the concept of controlling both the biological functionality and biomolecule release kinetics by means of polymeric blends with tunable degradation is very promising for biomedical applications in general, the outcome of this particular study represents a solid step for future research on PCL-blend-PEG coatings with respect to the loading and release of different bioactive molecules.

## 2. Materials and Methods

### 2.1. Materials

Polyethylene glycol (PEG, H(OCH_2_CH_2_)_n_OH, 6000 Da molecular weight) and polycaprolactone (PCL, (C_6_H_10_O_2_)_n_, 14,000 Da molecular weight) powders, both purchased from Sigma Aldrich (Darmstadt, Germany), were used in the coatings’ fabrication. Appropriate amounts of PCL and PEG (mixed in 3:1 and 1:3 weight ratios), as well as simple polymers, were dissolved in chloroform (Merck (Darmstadt, Germany), grade purity 99%), in a concentration of 30 g/L (w/v), prior to performing the dip-coating (DC) experiments. The polymeric coatings aimed for in vitro studies (degradation, electro-corrosion, and biocompatibility) were deposited on grade 4, commercial pure Ti disks (12 mm in diameter and 0.2 mm in thickness). For other investigations, 10 mm^2^ glass coverslips and IR transparent double-side polished Si (100) slides were used as deposition substrates.

### 2.2. Dip-Coating (DC) Deposition of PCL-Blend-PEG Coatings

Biodegradable coatings were deposited by immersing the substrates into the polymeric solution. The substrates were kept inside the solution for 1 min and then gradually pulled up at a withdrawal velocity of 100 mm/min, thus generating a layer on both sides of each substrate (Figure 1). The volatile solvent (chloroform) evaporates almost instantly, leaving just the polymeric coatings [32]. The coating thickness was evaluated by step profilometry using a Stylus Profiler XP-2 system (Ambios Technology, Santa Cruz, California, U.S.A.) working at a 1 mm/sec withdrawal speed.

The obtained thickness values range from 0.6 to 2 μm, depending on the composition of the polymer mixture. Coating thickness can be adjusted by varying the withdrawal speed, composition, or concentration of the deployed solution [32].

### 2.3. Physico-Chemical Characterization of PCL-Blend-PEG Coatings

Fourier transform infrared (FTIR) spectroscopy studies were conducted with a Shimadzu 8400S Spectrometer (Shimadzu Europa GmbH, Duisburg, Germany), in transmission mode and attenuated total reflection (ATR) using a diamond prism. The spectra were recorded within the range of 4000–550 cm^−1^, with 4 cm^−1^ resolution and 50 scans series per sample.

The crystallinity of deposited coatings was assessed by grazing incidence X-ray diffraction (GIXRD) using a Bruker D8 Advance diffractometer (Bruker AXS, Karlsruhe, Germany), equipped with a Cu target X-ray tube, in parallel beam setting. The incidence angle was set at 2°, while the scattered intensity was acquisitioned within the range of 10°–50° (2θ), with 0.04° step size and 6 s per step.

The surface wettability of the deposited coatings was evaluated by measuring the static contact angle (CA) for each coating at room temperature and expressed as the arithmetic average of three measurements ± SD (standard deviation). For this purpose, we used a drop shape analysis system, model DSA100 from (Kruss GmbH, Hamburg, Germany). Each sample was placed on a plane stage, under the tip of a water-dispensing disposable blunt-end stainless steel needle with an outer diameter of 0.5 mm. The needle was attached to a PC-controlled syringe pump ensuring water droplet delivery to the test surface. The volume of the water droplets was approximately 2 μL [33]. The dispensing of droplet as well as the CA analysis, were performed using the DSA3^®^ software supplied with the instrument. The CA value was estimated by fitting, either a second degree polynomial or a circle equation may be used to approximate the shape of the sessile drop. This was followed by the subsequent calculation of the tangent to the drop at the liquid–solid vapor interface. The camera was positioned to observe the droplet under an angle of about 2°–3° with respect to the plane of sample surface supporting the droplet [14]. The measurements were conducted in a constant temperature (25 °C) and relative humidity (50%) conditions.

### 2.4. Surface Morphology of PCL-Blend-PEG Coatings

The surface morphology of deposited coatings was investigated by scanning electron microscopy (SEM) using a FEI Inspect S scanning electron microscope (Thermo Fisher Scientific, Waltham, MA, USA). The investigations were performed at 10–20 kV acceleration voltage and 250–20,000 × magnification, in high vacuum. In order to reduce electrical charging, all samples were capped with a thin Au film prior to analysis. Compositional energy dispersive spectroscopy (EDS) analyses were performed with a SiLi type detector (model EDAX Inc., InspectS, Mahwah, NJ, U.S.A.), operated at 20 kV. The EDS analyses were conducted in duplicate on three distinct and nonoverlapping film regions having areas of 250 μm × 250 μm.

### 2.5. Electrochemical Behavior of PCL-Blend-PEG Coatings

The electrochemical behavior of simple and blend coatings was evaluated taking several approaches, as follows: (i) open circuit potential (*E_OC_*) for 1 h (the steady state was achieved), (ii) Tafel plots at ±250 mV vs. *E_OC_* with a scan rate of 1 mV/s, and (iii) potentiodynamic polarization curves from −1 V (vs *E_OC_*) to +1 V (vs. SCE) at 1 mV/s. All electrochemical tests were performed using a potentiostat/galvanostat (PARSTAT 4000, Princeton Applied Research - Ametek, Oak Ridge, TN, USA) coupled with a low current interface module (VersaSTAT LC, Princeton Applied Research - Ametek, Oak Ridge, TN, USA). A three electrodes configuration electrochemical cell was used to perform the tests. Namely, the saturated calomel electrode (SCE) served as reference, the platinum one for recording and the investigated sample as the working electrode, respectively. The samples were mounted on a Teflon support so that the tested surface had an exposed area of 0.5 cm^2^. Over the course of the electrochemical experiments, the electrochemical cell along with the low current interface module was inserted into a Faraday cage to eliminate the interference with electromagnetic fields. The tests were performed in SBF (pH = 7.4), at human body temperature 37 ± 0.2 °C using a CW-05G (Jeio Tech, Seoul, Korea) heating and recirculation bath model. Experiments were performed in triplicates for both substrate and coating settings, respectively. The obtained data are presented as mean ± SD.

### 2.6. Degradation Behavior of PCL-Blend-PEG Coatings

To simulate the processes occurring inside human tissues [34], the polymeric-coated metallic samples were tested under physiological-mimicking dynamic conditions (in SBF at 37 °C) using a manufactured set-up consisting of a multichannel degradation cell (calibrated hoses of 1.6 mm inner diameter) (Figure 2) [35]. The multichannel cell was connected to a peristaltic pump from (Ismatec Wertheim, Germany) and a thermostatic bath (Grant TC 120, Fisher Scientific Ltd., Vantaa, Finland). The flow rate was established at 1.31 ± 0.04 mL/min [36] through each channel. For the sake of statistics and reproducibility of the data collection approach, two samples were used for each channel and completely immersed in the channel’s degradation cell filled with 5 mL SBF prepared according to Kokubo’s recipe [34]. The weight measurements of specimens were acquired over different time laps of up to 120 min (for PEG-coated samples) and up to 16 weeks (in the case of PCL and PCL–PEG coatings). At the end of the degradation experiment, the samples were carefully rinsed with deionized water, dried in vacuum, and subsequently weighed. For all mass measurements (performed on either bare or coated titanium substrates, before and after degradation) a Partner Radwag Mya 0.8/3.3Y analytical microbalance with an accuracy of ± 3 μg was used [35].

The mass loss curves were plotted using the data obtained by evaluating the mass of each film, on each channel, before and after the considered dynamic testing periods. Furthermore, for the mass loss measurements, coated titanium substrates were weighed before and after degradation, respectively. The corresponding values were obtained by averaging three independent weight measurements performed in identical conditions. We labeled the mass before deposition as the mass of the substrate and the mass after deposition as the mass of substrate coated with polymer. Consequently, the mass of the deposited film is the difference between the above two values. In order to estimate weight variations produced during dynamic conditions, we calculated the mass difference between the initial and final mass of polymer coatings, measured before and after degradation, respectively. Meanwhile, the relative loss mass was calculated using the formula: [m_f_−m_i_)/m_i_] × 100%, where m_i_ and m_f_ represent the initial and final mass of polymer coating, before and after dynamic testing [30,31,32].

### 2.7. In Vitro Assays of PCL-Blend-PEG Coatings

Human mesenchymal stem cells (MSCs) were isolated from bone marrow aspirates [37]. The bone marrow was separated, by centrifugation on sterile medium Ficoll-Paque PLUS (GE Healthcare-Life science, Pittsburgh, PA, USA). Isolated cells were cultivated in low-glucose (1 g/L) Dulbecco’s Modified Eagle Medium (DMEM) supplemented with 10% fetal calf serum, 1% Glutamax, 50 U/mL penicillin, 50 mg/mL streptomycin (all from Invitrogen- Life Technologies, Paisley, UK) at 37 °C, in a humid atmosphere containing 5% CO_2_. The culture medium was changed every 3 days and cells were split after approximately 10 days.

To confirm the expression of specific markers, cells at passage 2 were immunophenotyped. All validated cultures were used for further experiments or cryopreserved.

Human aortic endothelial cells, line EA.hy926 (ECs) [38] were cultured in high-glucose (4.5 g/L) DMEM supplemented with 10% fetal bovine serum (EuroClone SpA, Pero (MI),Italy) and antibiotics (100 U/L penicillin, 100 U/L streptomycin, 50 U/L neomycin) at 37 °C, 5% CO_2_, in a relative humidity environment of over 95%. For cell viability assay and cell morphology examinations, cells at passage three and four were utilized.

For cell adhesion assay, the cytoskeleton architecture was analyzed by fluorescence microscopy using actin staining. MSCs were seeded on polymer-coated titanium disks at 5000 cells/cm^2^ density, while ECs were used at 10^5^ cells/mL density, in 24-well plates. Bare titanium disks were used as controls. After 9 days for MSCs and 3 days for ECs, respectively, the cells were washed and fixed in 4% p-formaldehyde (PFA) for 15 min at room temperature (RT). Cells were permeabilized using 0.2% Triton X-100 for 3 min and a 0.5% bovine serum albumin (BSA) solution was –used for blocking the nonspecific sites. The cells were washed and subsequently exposed for 30 min at RT to Alexa Fluor 488-conjugated Phalloidin (1:100) (Invitrogen Life Technologies, Paisley, UK), to label the actin cytoskeleton. Samples were mounted using Prolong Gold Antifade Reagent (Invitrogen, Life Technologies, Paisley, UK). Fluorescence images were acquired using a Zeiss Axio Imager Z1 with ApoTome Module (Berlin, Germany).

To assess potential cytotoxic effects of the materials on cell viability, the LIVE/DEAD Viability/Cytotoxicity assay Kit (Lonza Walkersville, Inc, MD, U.S.A.) was used. The method is based on the simultaneous determination of both live and dead cells by measuring two recognized parameters of cell viability: intracellular esterase activity of live cells detected with calcein AM and plasma membrane disruption detected by ethidium homodimer-1 incorporation into dead cells and binding to nucleic acids, respectively. After 3 days in culture, the medium was removed and the endothelial cells were incubated with 1 µM calcein AM and 2 µM ethidium homodimer-1, for 45 min. Both live and dead cells were analyzed by fluorescence microscopy (calcein AM ex/em-485/530nm, ethidium homodimer-1 ex/em-530/645nm and Filter Set 77, Axio Vert. A1 inverted microscope with Apotome, Zeiss. All chemicals were from Sigma-Aldrich (St Louis. MO, USA), unless otherwise specified.

## 3. Results

### 3.1. FTIR Investigations

Relevant data regarding the composition of PCL, PEG, and mixed coatings are included in the FTIR spectra presented in Figure 3.

The typical triplet of C–O–C stretching vibration (corresponding to skeletal H_2_C–O–CH_2_ within PEG) can be observed at 1148, 1112, and 1060 cm^−1^. The strong band centered at 1280 cm^−1^ is assigned to the overlapped twisting vibrations of CH_2_ (from PCL) and to the asymmetric stretching vibrations of C–O–H (from PEG) [39]. One notes the peaks of PEG at 960 and 843 cm^−1^ (corresponding to the symmetrical vibrations of C–O–C bonds) [40] and in the range of 1360–1470 cm^−1^ (corresponding to the C–H bending). According to the literature, rocking vibrations of CH_2_ can be noticed at 1360 cm^−1^, while deformation vibrations of CH_2_–C=O occur around 1420 cm^−1^ (both corresponding to PCL). However, within this wavenumber range, deformation vibrations of both O–C–H and C–O–H (from PEG) can overlap. Additionally, bending vibrations of the C–H group, (also corresponding to PEG) appear in the vicinity of 1360 cm^−1^. The 1729 cm^−1^ absorption maxima (C=O stretching vibrations of the ester carbonyl group within PCL) and the 1189 and 1240 cm^−1^ doublet (C–O–C stretching vibrations [41] of the repeated –OCH_2_CH_2_ units of PEG and –COO bonds stretching vibrations), both also observed by Wang et al. in their studies of PEG composites [41], can be further traced in Figure 3. This spectral area was also affected by the C–O stretching vibrations and C–H deformation vibrations, both issued from PEG. Correspondingly, PCL exhibited the 2943 cm^−1^ (–C–H asymmetric stretching), 2883 cm^−1^ (–C–H symmetric stretching) and 1727 cm^−1^ (–C=O stretching) characteristic peaks.

The FTIR spectra of PCL-blend-PEG exhibited the main characteristic peaks of both polymers, evidencing the retention of main functional groups within pristine organic precursors. The most important features are the presence of the intense absorption bands centered at 1725 cm^−1^ (evidence of stretching vibrations of the carbonyl groups within PCL) and at 1111 cm^−1^, respectively. The latter one is due to the strong vibrations of C–O–C ester structural units within PEG. Such features are an indicative of blend formation. In addition, the broad bands centered at 2945 cm^−1^ (corresponding to C–H stretching vibrations of CH_2_ within the PCL polymer), respectively, at 2868 cm^−1^ (attributed to C–H stretching vibration of the PEG polymer) evidence the presence of association forms between the functional groups of constituent polymers [40,42].

### 3.2. GIXRD Investigations

X-ray diffraction (GIXRD) patterns of dip-coated films given in Figure 4 have confirmed the presence of compounds in polycrystalline states. We note a sum-up of both PEG and PCL diffraction peaks, indicating the preservation of the crystalline nature after mixing. However, it is difficult to directly identify the characteristic peaks of the polymeric mix, since PEG peaks were partially overlapped with PCL peaks.

As evidenced in Figure 4, PEG exhibited a sharp peak at 2θ = 19.3° and a broad one at 23.2°, while PCL shows three different characteristic maxima, thus confirming its semicrystalline structure. The peaks of PCL found at 21.5°, 22.1°, and 23.8° are attributed to (110), (111), and (200) reflection planes of the orthorhombic crystal [43]. Similar crystallization behavior was reported for PCL networks morphologies [43] and for PCL-based composites [44]. Diffraction patterns of the physical mixture exhibit characteristic maxima of both of the basic polymers, the obtained results being in agreement with previously reported FTIR data.

### 3.3. Coatings Wettability

Estimated contact angles for the PCL-blend-PEG coatings exhibited smaller values as compared to those for the simple PCL-coated samples, but larger ones than those corresponding to the PEG-coated ones (Table 1).

Wettability measurements evidenced a pronounced hydrophilic character of the blend coatings, especially when increasing the amount of the PEG constituent. In fact, numerous literature reports have already established the direct connection between the overall enhancement in hydrophilicity of a coating and the addition of PEG in PEG-based composites, as also discussed by Fu et al. in [45].

It is known that the surface adsorption of water molecules occurs immediately after the insertion of an implantable material or device inside a biological medium [46]. Secondly, the interaction between the implant and the host organism relies on the surface absorption of the physiological proteins [46]. Furthermore, materials with moderately hydrophilic surfaces facilitate the adhesion and subsequent cell growth, ultimately leading to superior biocompatibility [46,47,48,49]. Figure 5 illustrates the hydrophilic behavior of our obtained blend coatings.

At this point, we can state that the overall hydrophilicity of the PCL-blend-PEG coatings is not only superior to the PCL ones but this hydrophilicity can be significantly improved by the presence of the PEG constituent. The diminishment in CA values, proportional with the increase of PEG content, is mainly due to the solubilization of PEG during the contact of the water droplets with the coating’s surface. Nevertheless, as we previously discussed, the addition of PEG alters the structural and morphological features of coatings. Further insights on this aspect are offered in the following sections.

### 3.4. Coating Degradation Behavior

The degradation behavior of PCL-blend-PEG coatings was evaluated by mass loss measurements, under long-term immersion in simulated conditions, as well as under electrochemically accelerated ones. Comparative microstructural investigations and associated interpretations are provided for both scenarios.

During our studies, we observed that the degradation rate of PCL-based coatings was progressive over the considered period of time (2 to 16 weeks) with the expected extreme behaviors shown in Figure 6.

The relative mass loss curves were plotted and fitted using the Origin® 2019 software (Figure 6). Polynomial curves fit the degradation of simple PCL, PCL-blend-PEG (3:1), and PCL-blend-PEG (1:3) coatings, while an exponential curve is appropriate for the simple PEG. Such degradation behavior is found in previous reports on block copolymers of PCL/PEG [50,51]. The degradation process was somewhat slower in the beginning but it became faster over time, leading to significant weight loss. This is most probably due to polymer swelling followed by a fragmentation of its chains and their subsequent diffusion. For instance, in the case of blend samples with the highest PCL content (i.e., PCL-blend-PEG (3:1)), the percentage in mass loss increased from 17% (after 2 weeks) to almost 82% (after 16 weeks). The simple PCL samples followed the same tendency, as expected. Furthermore, after an immersion in SBF of 4 weeks, the slowest degradation rate (of 4.5%) was exhibited by the simple PCL coatings. This rate further increased to 13.7% after 12 weeks. It can be therefore concluded that in the case of simple PCL and PCL-blend-PEG (3:1) coatings, the mass of polymers remains almost unchanged within the first period (2 to 4 weeks) of the degradation process. From this moment on, a rapid increase in the release of polymer mass occurs, due to accelerated diffusion of smaller polymer chains [9], as stated above.

As expected, PEG-coated samples exhibited the fastest solubility rate under biologically simulated dynamic conditions. The polymer release rate was 93%, after 48 min, the degradation being less pronounced for the rest of the dynamic testing. After 120 min, the release of PEG was considerably reduced, especially since, by this time, the aliphatic polyether was almost entirely solubilized (98%). It is known that the PCL aliphatic polyester has an intrinsic extended degradation time and its prolonged presence in the organism can delay tissue regeneration. Therefore, it was expected that the mixture between PCL and the highly soluble PEG would accelerate the degradation of resulted blends by increasing water uptake and accelerating the release of hydrophilic degradation products [52,53,54,55]. Our results are in good agreement with previously published data [56], supporting the evidence that the addition of PEG accelerates PCL degradation and thus, suggesting the possibility of modulating the degradation rate of blend patterns by using different concentrations of biopolymers. According to the literature, in various cases of PCL-block-PEG, specific factors, such as polymer molecular weight and composition, constituent polymer morphology, temperature, and pH, may significantly influence the degradation rate [57,58].

To further extend our degradation studies and for the sake of a better understanding of the mechanisms involved, we used SEM to observe the time-dependent morphological changes of either pristine or degraded polymeric blends. From a qualitative point of view, the EDS spectra of the deposited coatings (Appendix A) indicate the presence of typical elements only (C,O), along with the signal originating from the Ti substrate, the total absence of chlorine denoting that there is no contamination from the deployed solvent (chloroform).

As a general observation, the surfaces of all PEG-based coatings were quite flat before degradation (Figure 7, column 1). Significant changes in the morphology of PCL-containing coatings became visible after 8 weeks of dynamic exposure (Figure 7, column 2), with the surface appearing to erode extensively only after this moment on and closer to the 16 weeks’ timeline (Figure 7, column 3). For a better understanding, we present in Figure 8 SEM images of deposited samples before and after degradation, collected at higher magnification from selected area.

An alveolar morphology could be observed for all pristine PCL coatings. On a micrometric scale, the size and depth of these cavities tend to diminish in the case of blends. However, their number remained higher in PCL-blend-PEG (3:1) samples, dropping drastically with the increase in PEG content (PCL-blend-PEG (1:3)) (Figure 8). Following their immersion in SBF, a preferential degradation behavior was observed in the case of PCL and PCL-blend-PEG 3:1 samples; namely, connected regions with higher density of cavities became increasingly more visible on the surface (Figure 7). This behavior is strongly related to the PCL content and can be explained by the presence of the phase separations and inhomogeneities in the material density and possible results of the coatings’ synthesis. The long-term tests performed on the simple PCL polymer induced the appearance of holes (140 nm–3 μm in size) after 8 weeks. The diameter of these holes increased even further (240 nm–4.5 μm) during the remaining interval of up to 16 weeks.

It is noted that in the case of PCL-blend-PEG (3:1) samples, due to PEG addition, holes with greater diameter (i.e., at 8 weeks ~3.5 μm) could be observed. The accelerated degradation process was obvious, the diameters of the holes after 8 weeks being comparable with the ones that occur in the PCL samples around 16 weeks (~4 μm). Furthermore, for the PCL-blend-PEG (3:1) coatings, the appearance of cracks associated with partial delamination could be noticed in the vicinity of the larger holes after 16 weeks. In the case of PCL-blend-PEG (1:3) coatings, samples with fairly homogenous morphology were obtained, evidencing a better homogeneity of the polymeric blends. After 8 weeks of immersion in SBF of PCL-blend-PEG (1:3) samples, the morphology with holes and pits disappeared, being replaced by some aggregates with irregular shape, arbitrarily scattered on the surface. Similar aggregates could be observed also on the surface of PCL-blend-PEG (3:1) coatings. These polymeric particulates were in the micrometric range (260 nm–3.5 μm). After 16 weeks, polymeric agglomerations were still present on the surface of PCL-blend-PEG coatings, their distribution following the initial laminar morphology. At the same time, it should be stressed that the degradation occurred much faster due to the increased concentration of PEG.

SEM images of simple PEG coatings revealed very smooth films, with a more pronounced laminar morphology (Figure 7). At the end of the tests performed under dynamic conditions, significant changes regarding the structural and morphological integrity of simple PEG-coated surfaces could be observed. A similar morphological behavior for pristine PEG and PCL was reported also by others [59,60].

As a general remark, it is well known that interconnected pores and their corresponding size are considered key parameters for vascularization and cell migration. Therefore, the yielded sizes of degradation holes are encouraging for the bone regeneration and formation of capillaries [61,62].

FTIR spectra recorded by ATR (Appendix A) on PCL-based coatings, after degradation, exhibited similar characteristic bands as those of PCL pristine samples, irrespective of the degradation time interval. However, the infrared peaks of PEG rapidly disappeared over the time, especially in the case of samples with increased PEG content, due to its high solubility in the SBF solution. Therefore, we can say that residual products, resulted from degradation or solubilization, are released in the SBF solution and the remaining coating is mainly formed of PCL. We emphasize that PEG total elution is the main mechanism occurring during the first stage of degradation, followed by the erosive changes of blended material, process mainly due to the hydrolytic alteration of PCL that occurs slowly over the entire evaluation period. In agreement with the heretofore described degradation investigations, the electrochemical experiments used to induce accelerated degradation effects in identical immersion environments reveal similar behavior of the polymers and their blends over time. The potentiodynamic polarization curves corresponding to substrate and coatings, tested in SBF medium as electrolyte, are shown in Figure 9. For exemplification purposes, only one potentiodynamic polarization curve is illustrated for each sample.

Using the Tafel extrapolation, the main electrochemical parameters were extracted. These parameters, corrosion potential (*E_corr_*) and corrosion current density (*i_corr_*), characterized the electrochemical behavior of the investigated samples and are presented in Table 2.

The polarization resistance (*R_p_*) was calculated using the Stern–Geary equation [63]:(1)$$R_{p}=\frac{1}{2.3}\cdot \frac{\beta_{a}\cdot \left|\beta_{c}\right|}{\beta_{a}+\left|\beta_{c}\right|}\cdot \frac{1}{i_{corr}}$$
where the *i_corr_* is the corrosion current density, β_a_ - anodic slope, and β_c_ - cathodic slope of the Tafel plots.

These main electrochemical parameters serve in determining the protective efficiency coefficient (*P_e_*) of the coatings:(2)$$P_{e}=\left(1-\frac{i_{corr,\;coating}}{i_{corr,\;substrate}}\right)\cdot 100$$
where *i_corr,coating_* and *i_corr,substrate_* are the corrosion current densities of the coating and of the substrate, respectively.

Based on Elsener’s empirical equation (Equation (3)) [64] the porosity (P) is calculated as:(3)$$P=\left(\frac{R_{p,\;substrate}}{R_{p,\;coating}}\right)\cdot 10^{-\frac{\left|\Delta E_{corr}\right|}{\beta_{a}}}$$
where *R_p_*,_substrate_ and *R_p,coating_* are polarization resistance of the substrate and of the obtained coatings, respectively; Δ*E_corr_* is difference between the corrosion potentials of the coatings and of the uncoated substrate.

By comparing the *E_corr_* values, it can be observed that the PCL layer presented the most electropositive value (−292 ± 4.92 mV) and increasing the PCL concentration in the PCL-blend-PEG coatings led to an improvement in the electrochemical behavior of these layers (as PCL is more stable). Comparatively, the PEG-based layers with high PEG concentration had smaller electropositive values, the *E_corr_* corresponding to PCL-blend-PEG (1:3) blend approaching the one of bare Ti.

Considering the corrosion current density, the lowest value (41.06 ± 1.38 nA/cm^2^) was measured for the PCL layers, followed by those corresponding to the blends with higher PCL content (60.32 ± 1.27 nA/cm^2^), such coatings exhibiting a less pronounced electrochemical behavior compared to other films. The results are in good correlation with the previously presented degradation investigations. Current density values measured for simple PEG layers are similar to those occurring when corroding bare titanium substrates, but higher than the current densities corresponding to PCL-blend-PEG (1:3) polymer coatings. It is straightforward to conclude that the increase in the amount of PEG in blend formulations leads to higher i_corr_ values and, consequently, to a higher degradation rate.

These observations lead to an obvious highest R_p_ value of the PCL coatings, the value of the resistance decreasing with the increase in PEG content within the blends. The assessment of coatings in terms of protective efficiency confirm once again the suitability of PCL, its corresponding P_e_ having the highest value (54.28 ± 1.75%) as opposed to the PEG (7.10 ± 1.05). The accelerated electrochemical behavior of blended coatings reveals diminished values of the main electrochemical parameters when increasing the PEG content, confirming that the degradation rate of PCL-blend-PEG in SBF can be compositionally adjusted. It should be pointed out that, for all mentioned samples, the anodic peaks shifted toward an electropositive potential when increasing the PCL amount, indicating that the electrochemical processes are chemically reversible for these samples. This linear relationship is typical for redox-active polymers attached to electrodes and testifies to the stability of the synthesized samples [65,66]. Polymer degradation-related data (morphology and degradation kinetics) are in good agreement with the results obtained from the electrochemical tests. In addition, the SEM micrographs of the electrochemically degraded samples resemble those degraded over time when immersed in SBF (Figure 7).

A distinctive feature of the electrochemical corrosion is the increase in the holes’ diameter in the case of PCL-based coatings. The alveolar aspect of the coatings is preserved; however, the cavities seem to be larger, exhibiting an accelerated merger tendency. The higher the PCL percentage, the more pronounced are the cavities and a more pervasive tendency of holes’ unification is obvious. In contrast, a higher content of PEG in the blend leads to spherical formations on the coating’s surface as a consequence of the PEG’s high solubilization rate. Nevertheless, while the general degradation behaviors of all the coatings are similar, the electrochemical corrosion accelerates these behaviors as compared to the time degradation approach.

Our study offers a better understanding of the long-term degradation of PCL-blend-PEG coatings. PEG and PCL are immiscible polymers and they phase-separate in the solid state, as recently reported in the literature [67,68,69]. Therefore, in an early stage, after the PCL-blend-PEG’s immersion, the first released material is mainly identified as the water-soluble PEG component, while for long term immersion periods, residual products of insoluble CL oligomers result due to the slow degradation of the PCL [70]. It was previously reported that the PCL-blend-PEG degradation process is governed by the behavior of PCL, based on the polymer linkages cleavage [71,72]. This could happen either passively by hydrolysis of the ester linkages or actively by enzymatic reaction [73,74,75]. In the initial phase, free carboxylic groups catalyze the cleavage of remaining ester groups, then an autocatalytic process can be observed [29]. In the case of a random hydrolytic chain scission, this is followed by polymer weight loss due to the diffusion of small CL oligomers [29]. Electrochemical results revealed similar morphological changes but at a larger degradation scale due to the acceleration of the degradation mechanisms in the presence of the electrical field.

### 3.5. Biocompatibility of PCL-Blend-PEG Coatings

The biocompatibility of the polymeric blend coatings deposited by DC method was assessed by monitoring the viability and adhesion of endothelial cells (ECs) and mesenchymal stem cells (MSCs) grown on these surfaces.

Fluorescence microscopy images show a relatively uniform coverage of the coated Ti disks by ECs (Figure 10a–e), 3 days after cell seeding. Irrespective of film composition, the green staining shows that most of the cells were viable and only a small fraction (between 5%–10% as determined by LIVE/DEAD staining), corresponded to dead cells, as shown by detectedred fluorescence (arrows), (Figure 10a–e).

In vitro biocompatibility of EC cell adhesion performed on these substrates demonstrates an enhancement in the EC coverage on PCL-blend-PEG coated substrates as compared with PCL alone. The growth of the cells was not dramatically modified when seeded on PCL-blend-PEG coatings. However, the polymeric coverage offered an improved micro medium that increased cell density in spite of some apoptotic cells observed (between 5%–10% and marked with arrows), as shown in Figure 10.

Next, we assessed the adhesion and spreading of MSCs and ECs onto the polymeric coatings using fluorescence microscopy. Cells were investigated after 9 and 3 days, respectively, to allow for the coverage of the biomaterial surface, for both lineages.

The polymer coatings fabricated by DC did not interfere with cell spreading, as demonstrated by actin filaments morphology, which is comparable between cells cultured on polymeric coatings and on the titanium control sample (Figure 11). Figure 11 depicts fluorescent images of actin staining, showing optimal spreading with the analyzed substrates and the titanium control, as well as interconnections with neighboring cells. There is no evident trend of changes in cell morphology upon contact with the polymer coatings.

Both bone formation and regeneration of blood vessels ensure the success of implanted metallic devices, in terms of osseointegration and angiogenic functional recovery [66]. The presence of MSCs and ECs at the site of injury play key roles in osteogenic graft integration, enhanced infiltration of these cells ensuring early vascularization and bone formation [76]. The ability of dip-coated blend coatings to establish optimal interactions with both MSCs and ECs was herein demonstrated. The cells showed a healthy aspect, revealing a good adhesion and spreading on the blend coatings and no cytotoxic effect (Figure 10 and Figure 11). The PCL/PEG blend addition to the metallic support formed a coating that allowed the cells to resist and to form a linkage that will support the vascular network formation. Therefore, blending the hydrophobic PCL with PEG in adequate proportion (3:1) will promote a better and longer survival of the cells on the metallic implant, hence a faster route to vascular network formation. The behavior of ECs including cell morphology and viability was enhanced by the addition of PCL into PEG-based coatings, which also proved to be non- cytotoxic for the cells (both MSCs and ECs). The results support the idea that a better understanding of implant coatings degradation process will allow for more efficient metallic implant biofabrication. We highlight how the addition of PEG into PCL-based coatings results in striking changes of the physical properties (degradation, wettability, electrochemical behavior in SBF) and improved biological performance of the coated metallic implants. The DC technique unveils its potential for the fabrication of coatings based on PCL-blend-PEG, with customizable properties according to the specific requirements of the envisaged applications. Such blend polymer coatings offer a better control over the biodegradation behavior, providing superior physico-chemical and biological characteristics as compared to conventional titanium implants used for orthopedic applications.

## 4. Conclusions

SEM micrographs recorded for PCL and PCL-blend-PEG coatings immersed in SBF up to 16 weeks showed a degradation behavior dependent on the polymeric composition of the coatings. Moreover, the microstructural features of the pristine coatings influenced the surface modifications during the degradation processes that followed the immersion in SBF. In the case of PCL-based coatings, an alveolar morphology could be observed. The size and the number of the identified cavities varied with increase in the PEG content. Additionally, the samples with a higher PEG content unveiled a laminar morphology. After the immersion in SBF, samples with higher PCL content exhibited larger pits and holes, while the samples richer in PEG could be characterized by a more uniform erosion of the surface. The electrochemical results were in agreement with the data obtained during the immersion tests and provided an extended view on the behavior of blend coatings for long degradation periods. ATR FTIR spectra collected on the PCL-blend-PEG samples after 16 weeks of immersion in SBF showed that PCL was the only remaining constituent of the coatings. Fluorescence microscopy studies confirmed that these coatings are adequate substrates for the adhesion and spreading of both human mesenchymal stem cells (MSCs) and endothelial cells (ECs), preserving their phenotype.

## Figures and Tables

**Figure 1 polymers-12-00717-f001:**
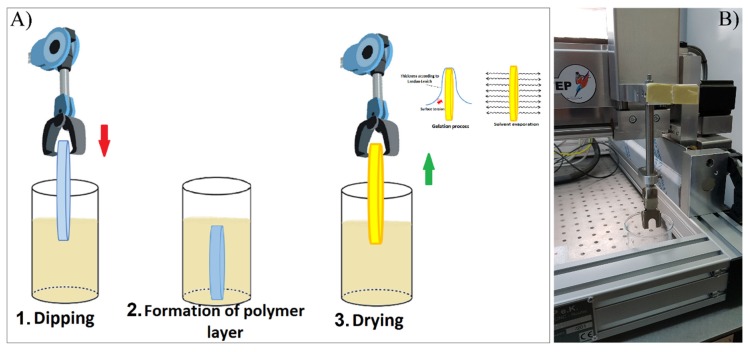
Dip-coating (DC) experimental set-up: (**A**) schematic description of the polymer deposition process: 1. dipping; 2. formation of the polymer layer; 3. drying. In this stage, the solvent evaporation occurs simultaneously with the gelation process, the thickness of the yielded coating being estimated according to the Landau–Levich theory. (**B**) Photograph of the laboratory set-up.

**Figure 2 polymers-12-00717-f002:**
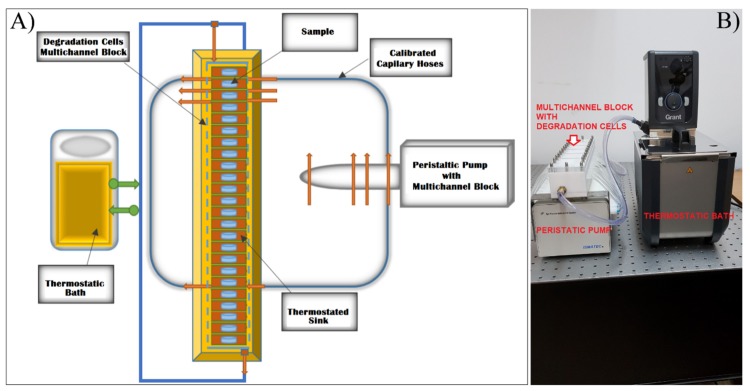
Degradation tests set-up: (**A**) schematic description; (**B**) photograph of the laboratory set-up.

**Figure 3 polymers-12-00717-f003:**
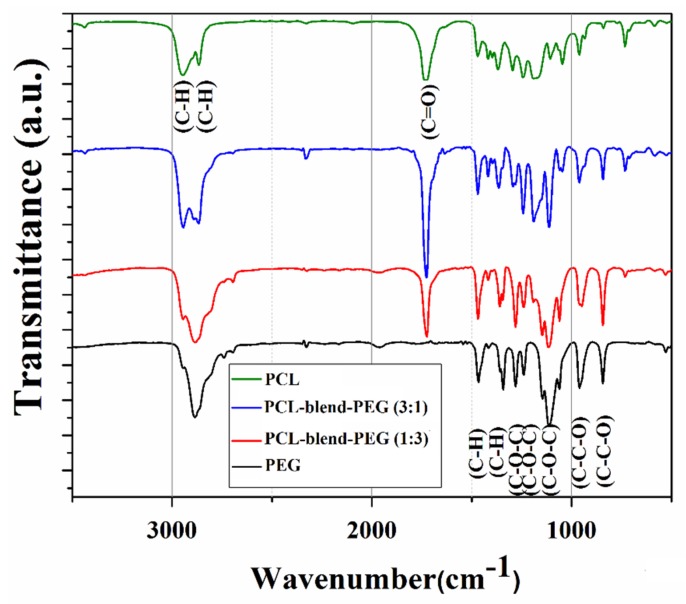
Typical FTIR spectra of dip-coated polymer films with polycaprolactone (PCL) (green); polyethylene glycol blends (PEG) (black); polycaprolactone-polyethylene glycol blends (PCL-blend-PEG) (1:3) (red); PCL-blend-PEG (3:1) (blue).

**Figure 4 polymers-12-00717-f004:**
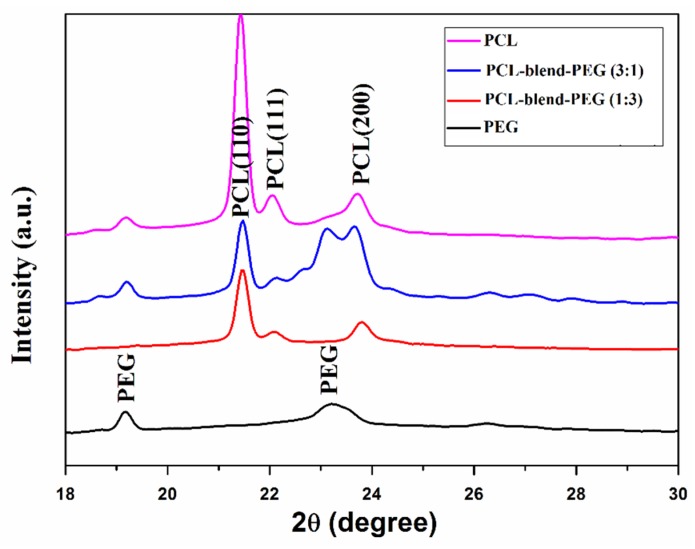
XRD diffractograms of dip-coated films: PCL (pink); PEG (black); PCL-blend-PEG (1:3) (red); PCL-blend-PEG (3:1) (blue).

**Figure 5 polymers-12-00717-f005:**
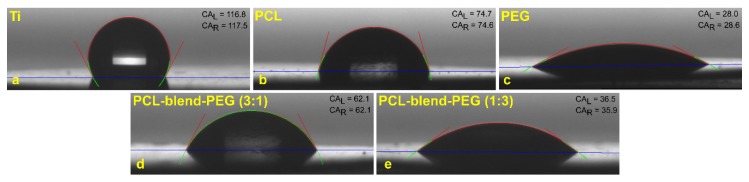
Typical images of water droplets on: (**a**) titanium substrates; (**b**) PCL; (**c**) PEG; (**d**) PCL-blend-PEG (3:1); and (**e**) PCL-blend-PEG (1:3). All contact angles are given in units of hexadecimal degrees. CA_L_ corresponds to the contact angle to the left side of the droplet while CA_R_ stands for the contact angle to the right side.

**Figure 6 polymers-12-00717-f006:**
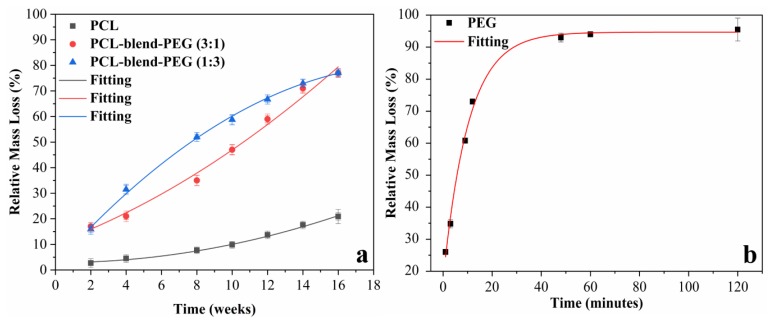
Relative mass loss curves illustrating the degradation behavior of polymeric coatings occurring during dynamic testing in SBF at 37 °C. The corresponding degradation time is 16 weeks for: simple PCL; PCL-blend-PEG (3:1), PCL-blend-PEG (1:3) coatings (**a**), and 120 min for simple PEG coatings (**b**), respectively.

**Figure 7 polymers-12-00717-f007:**
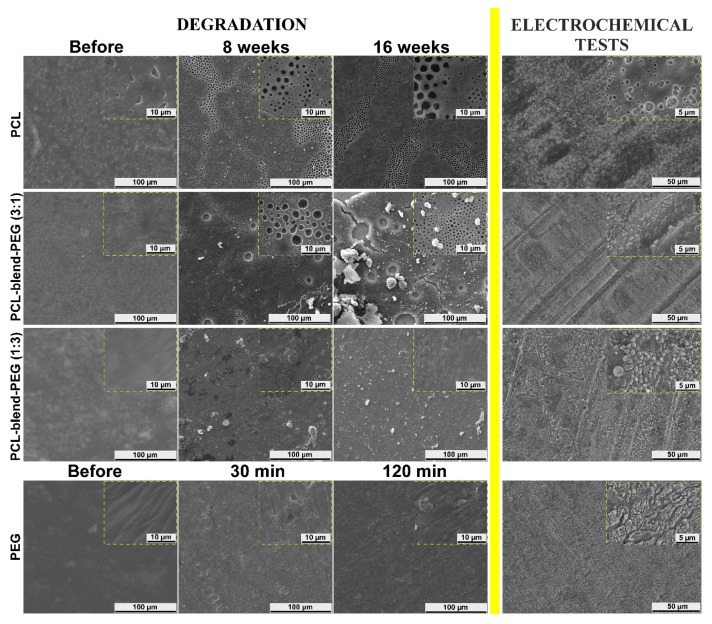
Representative SEM images of simple PCL and PCL-blend-PEG coatings depicted in the first panel, to the left of the yellow line: before degradation (column 1) and after degradation in SBF at 37 °C for 8 weeks (column 2) and for 16 weeks (column 3), respectively. The last row of the panel depicts simple PEG films before degradation (column 1) and after degradation in the SBF at 37 °C for 30 min (column 2) and for 120 min (column 3), respectively. Coatings’ morphology after electrochemical measurements in SBF is presented in the second panel, to the right of the yellow line.

**Figure 8 polymers-12-00717-f008:**
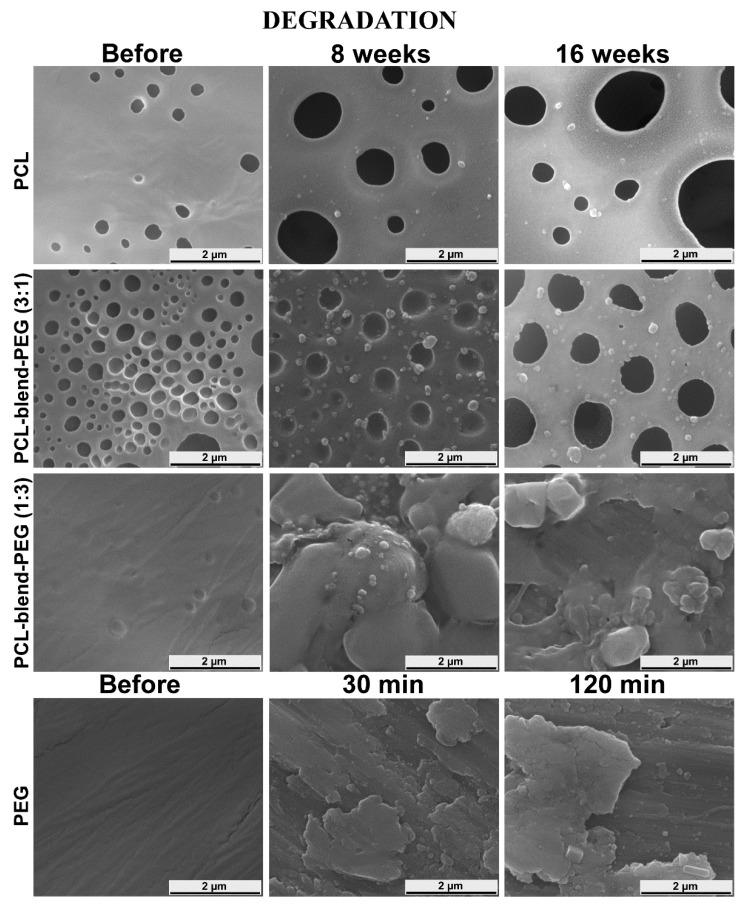
SEM images from selected areas of simple PCL and PCL-blend-PEG coatings: before degradation (column 1), after degradation in SBF at 37 °C for 8 weeks (column 2), and after degradation for 16 weeks (column 3), respectively. The last row of the panel depicts simple PEG films before degradation (column 1), after degradation in the SBF at 37 °C for 30 min (column 2), and after degradation for 120 min (column 3), respectively.

**Figure 9 polymers-12-00717-f009:**
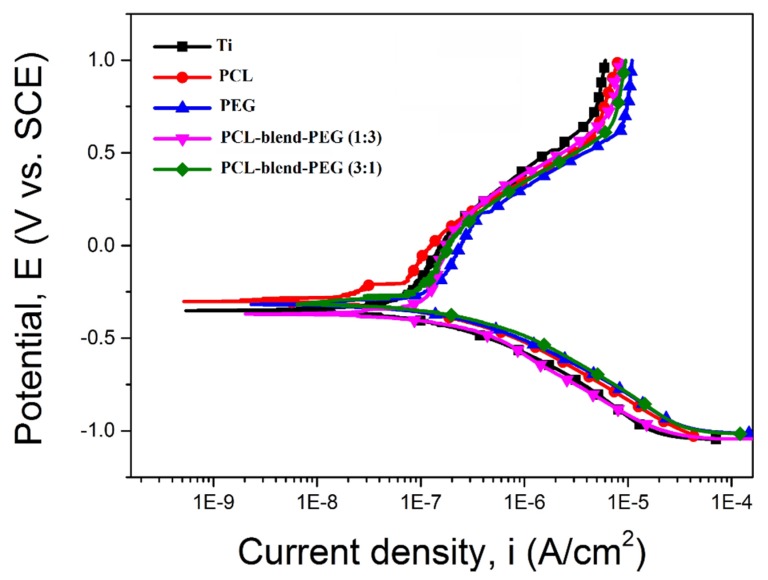
Potentiodynamic polarization curves of coatings and bare Ti substrate, respectively.

**Figure 10 polymers-12-00717-f010:**
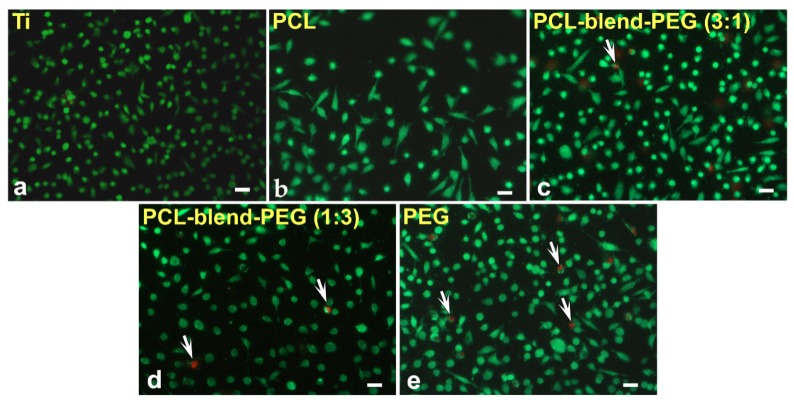
Cell viability assays of endothelial cells grown on PCL-blend-PEG coatings deposited by DC method. Living (green) and still attached dead (red) cells (marked with arrows) were detected by fluorescence microscopy using calcein AM and ethidium homodimer-1, respectively (20×; scale bar = 50 μm).

**Figure 11 polymers-12-00717-f011:**
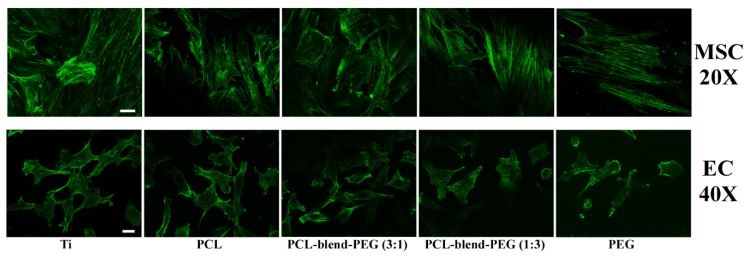
Fluorescence microscopy images of mesenchymal stem cells (upper panel) and endothelial cells (lower panel), adhesion after 9 and 3 days, respectively, seeded on PCL-blend-PEG coatings deposited by DC. Actin filaments were labelled with Alexa Fluor 488-conjugated Phalloidin (green) (upper panel 20×; scale bar = 50 μm; lower panel 40×; scale bar = 20 μm).

**Table 1 polymers-12-00717-t001:** Contact angle values for simple and blend coatings deposited by DC technique. (The data are given as mean ± SD).

Sample	PEG	PCL-Blend-PEG (1:3)	PCL-Blend-PEG (3:1)	PCL	Bare Titanium
Average CA values ± SD [degree]	28.3 ± 3.2	36.2 ± 0.9	62.1 ± 1.6	74.6 ± 4.3	117.1 ± 1.5

**Table 2 polymers-12-00717-t002:** Main electrochemical parameters of the electrochemical process (the data are presented as mean ± SD).

Sample	*E_corr_* (mV)	*i_corr_* (nA/cm^2^)	*R_p_* (ohm × cm^2^)	*P_e_* (%)	*P* (%)
Ti	−358 ± 8.04	87.25 ± 1.09	425.43 ± 11.68	NA	NA
PCL	−292 ± 4.92	41.06 ± 1.38	1169.85 ± 66.12	54.28 ± 1.75	30.33 ± 1.27
PEG	−317 ± 9.42	83.18 ± 1.81	552.72 ± 14.74	7.10 ± 1.05	72.95 ± 1.98
PCL-blend-PEG (1:3)	−356 ± 9.27	72.26 ± 1.05	486.02 ± 10.40	18.69 ± 0.94	82.35 ± 2.24
PCL-blend-PEG (3:1)	−307 ± 3.56	60.32 ± 1.27	620.50 ± 18.30	32.31 ± 1.45	60.85 ± 1.47

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
