# Peer review of "Long-Term Evaluation of Dip-Coated PCL-Blend-PEG Coatings in Simulated Conditions"

_polymers, 2020, doi:10.3390/polym12030717_

Round 1

Reviewer 1 Report

The reviewed manuscript presents the results of studies on degradation of the coating formed of poly (e-caprolactone) - blend - poly (ethylene oxide) applied by deep-coating method on a titanium surface  and the effect of this coating on the viability and adhesion of endothelial cells (ECs) and mesenchymal stem cells (MSCs). In the work, many advanced studies at a high level were carried out  unfortunately the manuscript presented has many gaps and errors in the methodology, which in the final result makes it difficult to draw reliable conclusions from the research. There is a lack of basic research that would explain the phenomena studied. In its current form, the work is highly speculative. The authors denote a blend of polycaprolactone with polyethylene glycol, by "PCL / PEG", which leads to confusion. In this way most often copolymers of polycaprolactone and PEG are determined (unformaly). I suggest using terminology in accordance with the IUPAC (Compendium of Polymer Terminology and Nomenclature IUPAC Recommendations 2008, p. 386) - poly (e-caprolactone) -blend-poly (ethylene oxide) Throughout the text, very often the authors refer to works that relate to block copolymers of polycaprolactone and PEG, not blends formed from polycaprolactone and PEG. This should be indicated in the appropriate places. This is particularly important in the descriptions of the degradation of such materials.

Line 91 - why PEG with a molecular weight of 6,000 was chosen in the research (whether it is  weight or number average mass, what is the true average molecular weight of this polymer). Describe what PEGs are biocompatible, in what molecular mass range. The choice of PEG type is very important because it involves the miscibility property of the blend as well as the dissolution rate of PEG in an aqueous environment.

Line 186 - why these cell lines were selected (MSCs, EA), why no tests were performed on bone cells (osteoblasts or chondrocytes)

Line 220 - there is no basic characteristic of the formed blende.

Part - 3.4. Coatings degradation behavior. Figure 6. It is more interesting to demonstrate the rate of degradation presented by the dependence of changes of ratio of the mass of released coat to the total initial coating mass in the time. The pictured graphs show that polycaprolactone degrades faster than the PCL-blend-PEG blend, which is rather impossible. If the leached PEG is released, how much of PCL remains on the surface?  We should observe a sharp setback of the mass loss course. Please describe how the blend composition changes with degradation time.

Figure 7 – Are the observed differences in the course of changes during degradation not caused to a large extent by the non-homogeneity of the obtained coating? During degradation, PEG leaches (quite easily soluble compound in water), which creates a porous structure and drives surface erosion processes. In the presented photos this effect it is hardly visible, too low resolution (you can repeat the SEM at a larger magnification and also  get the stereoscopic effect that allows you to assess the formation of craters and pits).

Part - 3.5. Biocompatibility of PCL / PEG coatings. Which of the presented  surfaces is the best ??? Nothing follows from the description. A more detailed commentary of the presented results is needed, it seems that the best results were obtained for PCL and Ti surfaces (in the case of EA cells), and very similar for Ti, PCL, PCL / PEG (3: 1) in the case of MSCs cells.

Part Discussion - Please describe how looks the improve of the biological properties of the coated surface, because from the presented results of biological research this fact does not follow.

Conclusion - the conclusions presented are not clearly explainable  by the results presented earlier. The purposefulness of covering the titanium surface with a polycaprolactone and PEG blend in the light of the presented results is not very convincing.

 For the above reasons, I do not recommend publication in its current form for publication.

Author Response

We appreciate the comments received from the referees, as they gave us the opportunity to further improve the quality of our manuscript. We amended the previous version of the manuscript in order to meet the received recommendations and suggestions. The changes made in the revised manuscript appear marked with “Track Changes” for an easy identification.

Please find below our detailed responses (point by point) to the comments:

In response to recommendations and suggestions of Reviewer 1:

Comment 1: The reviewed manuscript presents the results of studies on degradation of the coating formed of poly (e-caprolactone) - blend - poly (ethylene oxide) applied by deep-coating method on a titanium surface  and the effect of this coating on the viability and adhesion of endothelial cells (ECs) and mesenchymal stem cells (MSCs). In the work, many advanced studies at a high level were carried out  unfortunately the manuscript presented has many gaps and errors in the methodology, which in the final result makes it difficult to draw reliable conclusions from the research. There is a lack of basic research that would explain the phenomena studied. In its current form, the work is highly speculative. The authors denote a blend of polycaprolactone with polyethylene glycol, by "PCL / PEG", which leads to confusion. In this way most often copolymers of polycaprolactone and PEG are determined (unformaly). I suggest using terminology in accordance with the IUPAC (Compendium of Polymer Terminology and Nomenclature IUPAC Recommendations 2008, p. 386) - poly (e-caprolactone) -blend-poly (ethylene glycol) Throughout the text, very often the authors refer to works that relate to block copolymers of polycaprolactone and PEG, not blends formed from polycaprolactone and PEG. This should be indicated in the appropriate places. This is particularly important in the descriptions of the degradation of such materials.

Answer 1:

We agree with the Referee that the mixture type (blend or block) between the two polymers influences the morphology and, further, degradation of the composite. To more clarity on this aspect, as Referee indicated, we adapted the whole manuscript. We have modified the terminology according to IUPAC (by replacing "PCL / PEG" with poly (e-caprolactone) -blend-poly (ethylene glycol)) all over the manuscript. There was also a subsequent notation of this designation (as PCL-blend-PEG). Moreover, we modificate accordingly the legends of all figures in the text.

Also, we mentioned throughout the text when we refered to studies related to block copolymers of PCL and PEG, especially in Results Section.

Comment 2: Line 91 - why PEG with a molecular weight of 6,000 was chosen in the research (whether it is  weight or number average mass, what is the true average molecular weight of this polymer). Describe what PEGs are biocompatible, in what molecular mass range. The choice of PEG type is very important because it involves the miscibility property of the blend as well as the dissolution rate of PEG in an aqueous environment.

Answer 2:

We agree with the Referee’s comment that the molecular weight influences the biocompatibility of the biopolymer. Poly(ethylene glycol) type used is 6000 Da, average molecular weight, suitable for biological application conform to provider indications (Sigma Aldrich, code 81260). According to literature, the PEG biocompatibility directly increases with the molecular weight but this implies also a decrease in the polymer solubility. According to our experience with PEG [A.Visan,.et al.,G.Socol,(2017)Antimicrobial polycaprolactone/polyethylene glycol embedded lysozyme coatings of Ti implants for osteoblast functional properties in tissue engineering, Applied Surface Science,417, 234-243]; [Grumezescu, V., et al., Socol, G. (2017). Laser deposition of poly(3-hydroxybutyric acid-co-3-hydroxyvaleric acid) – lysozyme microspheres based coatings with anti-microbial properties. International Journal of Pharmaceutics, 521(1-2), 184–195], in our study we made the convenient compromise in order to obtain a composite that is both biocompatible soluble/miscible in the solvent and susceptible to solubilization/degradation processes in biological environments.

Comment 3: Line 186 - why these cell lines were selected (MSCs, EA), why no tests were performed on bone cells (osteoblasts or chondrocytes)

Answer 3:

These cells have important function for bone healing and regeneration (MSCs), and they are responsible for vascularization of injured tissue (EA) and generation of bone tissue.

The aim of our study, as stated in the paper, was to assess the interaction between cells involved in bone regeneration and the proposed coatings. Mesenchymal stem cells residing in the bone marrow are the precursors of osteoblasts thereby bearing a key role in the new bone formation process and implants osseointegration. Endothelial cells support these processes by generating new blood vessels at the site of tissue regeneration. In order to better support our approach, we added a new phrase in Introduction Section, last paragraph, lines 86-88,  page 2) which sounds like:” Endothelial cells, which underline the inner surface of the vasculature, play a key role in angiogenesis, and therefore their interaction with titanium surfaces is an important factor influencing tissue healing.” As Referee suggested, we decided to strenghten / reinforce the importance of the chosen two cell lines by adding a second detailed phrase in the Section 3.5, first paragraph, 567-568 lines, page 20 which sounds like:

The presence of MSCs and ECs at the site of injury play key roles in osteogenic graft integration – enhanced infiltration of these cells ensures early vascularization and bone formation

Also a new reference (number 76) (Section 3.5, first Paragraph, Page20, line 568) in which the importance of vascularization in the process of new bone formation is highlighted: Harvestine et al., Sci. Adv. 2020; 6 : eaay2387:

We updated the references list accordingly.

Comment 4:Line 220 - there is no basic characteristic of the formed blende.

Answer 4: According to the Referee’s suggestion, we modified the phrase from Section 3.1., first paragraph, page 7. The new sentence sounds like

“Relevant data regarding the composition of PCL, PEG and mixed coatings are included in the FTIR spectra presented in Figure 3.”

Comment 5:Part - 3.4. Coatings degradation behavior. Figure 6. It is more interesting to demonstrate the rate of degradation presented by the dependence of changes of ratio of the mass of released coat to the total initial coating mass in the time. The pictured graphs show that polycaprolactone degrades faster than the PCL-blend-PEG blend, which is rather impossible. If the leached PEG is released, how much of PCL remains on the surface?  We should observe a sharp setback of the mass loss course. Please describe how the blend composition changes with degradation time.

Answer 5:

The Fig. 6, referring to mass loss measurements, from the section 3.4. Coatings degradation behavior  was modified, for a clearer understanding of the obtained results. According to the Referee suggestion we represented the degradation percentage (%) vs time, where degradation percentage represents (relative released mass loss weight/initial weight)*100).

Figure 6. Relative mass loss curves illustrating the degradation behavior of polymeric coatings occurring during dynamic testing in SBF at 37°C. The corresponding degradation time is 16 weeks for: simple PCL (a); PCL-blend-PEG (3:1)(c) and PCL-blend-PEG (1:3)(d) coatings, and 120 minutes for simple PEG coatings(b), respectively.

The differences reported by the reviewer may be due to the fact that although the deposition process was performed in same conditions (withdrawal velocity, concentrations), the weight of the transferred material can vary depending on the polymer / mixture (including molecular mass). There are weight variations between coatings, mainly because of their distinctive composition and secondly because of different polymers’ properties (e.g. molecular weight, viscosity, etc). It is worth mentioning that the mass of PCL coatings was larger than other. Regarding the blend composition changes with degradation time, FTIR spectra recorded by ATR , after degradation, on PCL-based coatings exhibit similar characteristic bands as those of PCL pristine samples, irrespective of the degradation time interval. However, the infrared peaks of PEG rapidly disappear over the time, especially in the case of samples with increased PEG content, due to its high solubility in the SBF solution. Therefore, we can say that residual products resulted from degradation or solubilization are released in the SBF solution and the remaining coating is mainly formed of PCL.

Comment 6:Figure 7 – Are the observed differences in the course of changes during degradation not caused to a large extent by the non-homogeneity of the obtained coating? During degradation, PEG leaches (quite easily soluble compound in water), which creates a porous structure and drives surface erosion processes. In the presented photos this effect it is hardly visible, too low resolution (you can repeat the SEM at a larger magnification and also  get the stereoscopic effect that allows you to assess the formation of craters and pits).

Answer 6:  For a better view, we added a new SEM image at higher magnification (Figure 8) and we insert new clarifying phrases in Section 3.4. Coatings degradation behavior, page 12, lines 379-380; page 13: lines 392-400; page 14,lines:414-419. The new inserted text sounds like:

As a general observation, the surfaces of all PEG-based coatings are quite flat before degradation (Figure 7, column 1). Significant changes in the morphology of PCL-containing coatings become visible after 8 weeks under dynamic exposure (Figure 7, column 2) the surface appearing to erode extensively only after this moment on and closer to the 16 weeks’ timeline (Figure 7, column 3). For a better understanding, we present in Figure 8 SEM images of deposited samples before and after degradation collected at higher magnification from selected area.

In particular, in the case of pristine PCL coatings, an alveolar morphology was observed.  At micrometric scale, the size and depth of cavities decrease for the pristine PCL-blend-PEG samples. In addition, their number increases for the PCL-blend-PEG (3:1) samples and then drastically decreases with the increase of PEG content (PCL-blend-PEG (1:3)) (Figure 8). Following their immersion in SBF, a preferential degradation behavior is observed in the case of PCL and PCL-blend-PEG 3:1 samples, namely connected regions with higher density of cavities are visible on the surface (Figure 7). This behavior is strongly related to the PCL content and can be explained by the presence of the phase separations and inhomogeneities in the material density which might result from the coatings’ synthesis. The long-term tests on the simple PCL polymer induced the appearance of holes (140 nm – 3 μm in size) after 8 weeks. The diameter of these holes increased even further (240 nm – 4.5 μm) during the remaining interval of up to 16 weeks.

It is noted that in the case of PCL-blend-PEG (3:1) samples, due to PEG addition, holes with greater diameter (i.e. at 8 weeks ~ 3.5 μm) were visible. The accelerated degradation process is obvious, the diameters of the holes (up to 3.5 μm) after 8 weeks being comparable with the ones that occur in the PCL samples around 16 weeks (~ 4 μm). Further, for the PCL-blend-PEG (3:1) blend coatings, the appearance of cracks associated with partial delamination can be noticed in the vicinity of the more larger holes after 16 weeks. In the case of PCL-blend-PEG (1:3) coatings, samples with fairly homogenous morphology were obtained, evidencing a better homogeneity of polymeric blends. After 8 weeks of immersion in SBF of PCL-blend-PEG (1:3) samples, the morphology with holes and pits disappears and is replaced by the appearance of some aggregates with irregular shape arbitrarily scattered on the surface. Similar aggregates are observed also on the surface of PCL-blend-PEG (3:1) coatings. These polymeric particulates are in the micrometric range (260 nm – 3.5 μm). After 16 weeks, polymeric agglomerations were still present with a distribution that follows the morphology of initial laminar morphology of the pristine PCL-blend-PEG (1:3) coatings. At the same time, it should be stressed that the degradation occurs much faster due to the increased concentration of PEG.

SEM images of simple PEG coatings obtained by DC reveal very smooth films, with a more pronounced laminar morphology (Figure 7). At the end of the tests performed under dynamic conditions, significant changes regarding the structural and morphological integrity of simple PEG coated surfaces may be observed.

Figure 8.  SEM images from selected area of simple PCL and PCL-blend-PEG blend films: before degradation (column 1), and after degradation in SBF at 37â—¦C for 8 weeks (column 2) and for 16 weeks (column 3), respectively. The last row of the panel depicts simple PEG films before biodegradation (column 1) and after degradation in the SBF at 37â—¦C for 30 minutes (column 2) and 120 minutes (column 3), respectively. 

Comment 7:Part - 3.5. Biocompatibility of PCL / PEG coatings. Which of the presented  surfaces is the best ??? Nothing follows from the description. A more detailed commentary of the presented results is needed, it seems that the best results were obtained for PCL and Ti surfaces (in the case of EA cells), and very similar for Ti, PCL, PCL / PEG (3: 1) in the case of MSCs cells. Part Discussion - Please describe how looks the improve of the biological properties of the coated surface, because from the presented results of biological research this fact does not follow.

Answer 7: The aim of our cellular assays was to evaluate potential negative effects of the proposed polymer coatings onto the viability and adhesion capacity of cell types involved in bone regeneration. The results have shown no evidence of materials cytotoxicity. Also, the cells have shown no morphological modifications (which would represent an indicator of incompatibility of the proposed substrates with MSCs and ECs adhesion). This information was clearly stated in the manuscript and demonstrated by the representative fluorescence microscopy fields of view:

The ability of dip-coated blend coatings to establish optimal interactions with both MSCs and ECs was herein demonstrated. The cells show a healthy aspect, revealing a good adhesion and spreading on the blend coatings and no cytotoxic effect (Figure 9, 10).”

Our assays do not have a quantitative approach to compare the materials performance. The purpose of the experiments was to assess if the dip-coated PCL-blend-PEG represent potential candidates to be used in implant coating.

We accepted the reviewer’s opinion and introduced supplementary paragraphs for a more detailed commentary of the presented results on cell behavior in contact with biomaterials:

Section 3.5., first paragraph, page 19, lines 546-550

In vitro biocompatibility of EC cell adhesion performed on these substrates demonstrated enhanced of EC coverage on PCL-PEG compared with Ti or PCL alone. The growth of the cells is not dramatically modified when seeded on PCL/PEG blend coating. However, the polymeric coverage offers an improved micro medium that stimulate the cell proliferation in spite of some apoptotic cells observed (between 5%-10% marked with arrows) as shown in Figure 10.

Section 3.5., first two paragraphs, page 20, lines 571-577

The PCL/PEG blend addition to the metallic support formed a coating that allowed the cells to resist and to form a linkage that will support the vascular network formation.

The behavior of ECs including cell morphology and viability were enhanced by the addition of PCL into PEG-based coatings, that in not cytotoxic for the cells (both MSCs and ECs). The blending of hydrophobic PCL with PEG in adequate proportion (3:1) will promote a better and longer survival of the cells on the metallic implant.

Comment 8:Conclusion - the conclusions presented are not clearly explainable  by the results presented earlier. The purposefulness of covering the titanium surface with a polycaprolactone and PEG blend in the light of the presented results is not very convincing. For the above reasons, I do not recommend publication in its current form for publication.

Answer 8: We redistribute some parts of Discussion Section through the text (At the end of Section 3.5, page 20, lines: 578-589) and we remove it from the structure of the manuscript. 

We also rewrote the Conclusion Section, for better emphasizing the obtained results. Now, Conclusion Section sounds like.

  1. Conclusions

SEM micrographs recorded for PCL and PCL-blend-PEG coatings immersed in SBF up to 16 weeks showed a degradation behavior dependent on the polymer composition of the coatings. Moreover, the microstructural features of the pristine coatings influenced surface modifications during degradation processes occurred following the immersion in SBF. In the case of PCL-based coatings an alveolar morphology is observed. The size and number of the identified cavities vary with increase of the PEG content. Additionally, the samples with a higher PEG content unveiled a laminar morphology. After immersion in SBF, in the case of samples with higher PCL content larger pits and holes are observed, while for samples with higher content of PEG a more uniform erosion of the surface is obtained. The electrochemical results are in agreement with the data obtained during the immersion tests and provide an extended view on the behavior of blend coatings for long degradation periods. ATR FTIR spectra collected on the PCL-blend-PEG samples collected after immersion in SBF after 16 weeks showed that PCL is the only constituent of the coatings. Fluorescence microscopy studies confirm that these coatings are adequate substrates for the adhesion and spreading of both human mesenchymal stem cells (MSCs) and endothelial cells (ECs), preserving their phenotype.

Reviewer 2 Report

It is routine contribution devoted to coatings based on polycaprolactone-polyethylene glycol (PCL/PEG) blends. The coatings were prepared from chloroform solutions. However, it is well known that chloroform has a severe toxicity and high environmental impact. Thus, its use clearly counteracts the sustainability principles. It would be therefore to determine the distribution/quantity of chlorine in the coating using energy-dispersive X-ray spectroscopy (EDS, comp. e.g.: DOI: 10.1021/acs.biomac.7b01636).

Author Response

In response to recommendations and suggestions of Reviewer 2:

We would like to thank to the Referee for the interesting comments and useful suggestions, which helped us improve the manuscript.

Comment 1: It is routine contribution devoted to coatings based on polycaprolactone-polyethylene glycol (PCL/PEG) blends. The coatings were prepared from chloroform solutions. However, it is well known that chloroform has a severe toxicity and high environmental impact. Thus, its use clearly counteracts the sustainability principles. It would be therefore to determine the distribution/quantity of chlorine in the coating using energy-dispersive X-ray spectroscopy (EDS, comp. e.g.: DOI: 10.1021/acs.biomac.7b01636).

Answer 1: The referrer's concern is justified but the used solvent is highly volatile and our assumption is that during the dipping process the chloroform evaporated rapidly. For convenience, at Referee recommendation we performed EDS analyses on deposited samples, on 3 different areas on the sample. As evidenced from the recorded images, presented below,chlorine doesn’t appear on the spectra. We introduced in the manuscript text another 2 phrases in :Section 2.4, page5, first paragraph, 150-153 lines.

„Compositional energy dispersive spectroscopy (EDS) analyses were performed with a SiLi type detector (model EDAX Inc., InspectS), operated at 20 kV. The EDS analyses were conducted in duplicate on three film regions having areas of 250 μm × 250 μm.”

And in Section 3.4, page12, third paragraph, 371-374 lines.

„From a qualitative point of view, the EDS spectra of the deposited coatings (data not shown) indicated the presence of typical elements only (C,O), along with the signal originating from the Ti substrate and absence of chlorine being evidenced that no contamination of the used solvent (chloroform) occurs.”

The figure of EDS spectra of the deposited coatings is presented lower:

The EDS spectra of the deposited coatings

Reviewer 3 Report

Title: Long-term evaluation of dip-coated PCL/PEG coatings in simulated conditions

This research work concentrates on the long-term degradation under simulated conditions of bioactive coatings based on different compositions PCL/PEG blends fabricated for titanium implants by dip-coating technique.

Good English was used for this paper and this manuscript written well which sound scientifically.

Interestingly the authors showed the biological assessment which unveils the beneficial influence of PCL/PEG coatings for the adhesion and spreading of both human-derived mesenchymal stem cells and endothelial cells.

Minor corrections:

Figure 1 caption – Does not make sense. Please re-write this in a more clear format. Figure 2 caption – Does not make sense. Please re-write this in a more clear format. Figure 4 caption – Need more explanation of the graph. Table 1 - All numbers need unit for example 28.3±3.2 what is unit for 28.3 and what is unit for 3.2 (Standard deviation :SD or Standard error: SE) Figure 5 – Labelling and figure chapter need to be corrected. Need better labelling. Figure 6 – Labelling and figure chapter need to be corrected. Need better labelling. Table 2 - All numbers need unit for example 28.3±3.2 what is unit for 28.3 and what is unit for 3.2 (Standard deviation :SD or Standard error: SE) Figure 9 – What arrows represents? Need to explain in caption.

Overall, interesting paper with good results. I accept with minor corrections.

Author Response

In response to recommendations and suggestions of Reviewer3:

We are grateful to the Reviewer for the valuable and constructive remarks. We took into consideration all of them in the revision of the manuscript. Our list follows:

Comment 1:

Title: Long-term evaluation of dip-coated PCL/PEG coatings in simulated conditions

This research work concentrates on the long-term degradation under simulated conditions of bioactive coatings based on different compositions PCL/PEG blends fabricated for titanium implants by dip-coating technique.

Good English was used for this paper and this manuscript written well which sound scientifically.

Interestingly the authors showed the biological assessment which unveils the beneficial influence of PCL/PEG coatings for the adhesion and spreading of both human-derived mesenchymal stem cells and endothelial cells.

Minor corrections:

Figure 1 caption – Does not make sense. Please re-write this in a more clear format.

Answer 1: As Refferee suggested, we modified Figure 1 and its caption from” Figure 1. Schematic representation (A) and the photo (B) of dip-coating (DC) experimental set-up ” to „Figure 1. Dip-coating (DC) experimental set-up: Schematic description of the polymer deposition process: 1. Dipping; 2.Formation of polymer layer through a gelation process according to the Landaou-Levich Theory; Drying. (A) and the photograph of the laboratory installation (B)

Comment 2: Figure 2 caption – Does not make sense. Please re-write this in a more clear format.

Answer 2: For a better understanding, we modified Figure 2. and its Caption from: „Manufactured multichannel set-up for coatings degradation tests – schematic representation (A) and real image of experimental installation (B)” to : „Figure 2. Degradation tests set-up for coatings : schematic description (A) and the photograph of the laboratory installation (B)

 Comment 3: Figure 4 caption – Need more explanation of the graph.

Answer 3: We completed Figure 4 caption with more details. Now, Figure 4 caption sounds like:” Figure 4. XRD diffractograms of dip coated films: PCL (pink); PEG (black); PCL-blend-PEG (1:3) (red); PCL-blend-PEG (3:1) (blue)”

Comment 4: Table 1 - All numbers need unit: for example 28.3±3.2 what is unit for 28.3 and what is unit for 3.2 (Standard deviation :SD or Standard error: SE)

Answer 4: We added the required completions in Table I (Section 3.3. Coatings wettability; page 10, starting line 294-295)

Comment 5: Figure 5 – Labelling and figure chapter need to be corrected. Need better labelling.

Answer 5: We modified Figure 5 and consequently the figure 5 caption, that sounds now like: “Figure 5. Typical images of water droplets on:  titanium substrates(a);  PCL(b);PEG(c);PCL-blend-PEG (3:1)(d)and PCL-blend-PEG (1:3)(e). All contact angles are given in units of hexadecimal degrees. CAL corresponds to the contact angle to the left side of the droplet while CAR stands for the contact angle from the right side.

Comment 6: Figure 6 – Labelling and figure chapter need to be corrected. Need better labelling.

Answer 6: We modified figure 6 and corrected the caption from :” Figure 6. Mass loss curves illustrating the degradation behavior occurring during dynamic testing in SBF at 37°C. The corresponding degradation time is 16 weeks for simple PCL and PCL/PEG blend coatings, and 120 minutes for simple PEG coatings, respectively.” to :” Figure 6. Relative mass loss curves illustrating the degradation behavior of polymeric coatings occurring during dynamic testing in SBF at 37°C. The corresponding degradation time is 16 weeks for: simple PCL (a); PCL-blend-PEG (3:1)(c) and PCL-blend-PEG (1:3)(d) coatings, and 120 minutes for simple PEG coatings(b), respectively.

Comment 7: Table 2 - All numbers need unit for example 28.3±3.2 what is unit for 28.3 and what is unit for 3.2 (Standard deviation :SD or Standard error: SE)

Answer 7: We added all units for numbers in Table II (page 17, starting with line 478-479) in the text. For a clearer understanding,we pointed out that: “The data were presented as mean ± SD” on Section 2.5., page 5, line 170.

Comment 8:  Figure 9 – What arrows represents? Need to explain in caption.  Overall, interesting paper with good results. I accept with minor corrections.

Answer 8:

Arrows in figure 9 represent the few ethidium homodimer stained dead cells present on the surface of the analyzed samples. The figure caption has been modified to include this information:

Figure 9. Cell viability assays of endothelial cells grown on PCL/PEG blend coatings deposited by DC method. Living (green) and still attached dead (red) cells (marked with arrows) were detected by fluorescence microscopy using calcein AM and ethidium homodimer-1, respectively (20×; scale bar = 50μm).

Round 2

Reviewer 1 Report

I reviewed this work before, I advised it to be rejected. This decision was associated with the need to make a large number of changes in the presented manuscript.

The current version contains the expected changes, mostly regarding my previous comments. The text  is thoroughly reworded and supplemented with new data. Additional drawings have also been introduced. In this actual form, it seems to me that this work has become interesting and suitable for publication. Only relatively small patches remain to be done. Autors should thoroughly review the text and remove errors such as, for example;

- used expressions such as; "PCL-blend-PEG blend films", should rather be "PCL-blend-PEG films" such errors appear, for example, in the description of  figure 7 and 8.

- Figure 6,   in order to better understand the results of coating degradation process, autors shoul to establish the same scale and range in the relevant drawings (max mass loss 100%, max time 16 weeks), or place all functions (curves) on the one chart.

It seems to me also, that they should check whether the observed phenomenon of faster degradation of the coating made from the blend compared to the rate of degradation of the poly(e-caprolactone) coating is not associated with the practical total elution of PEG in the first phase of degradation, and then erosive changes arising on this way  was resulting in observed phenomenon of acceleration.

Author Response

In response to recommendations and suggestions of Reviewer 1:

We appreciate the comments received from the Referee, as it gave us the opportunity to further improve the quality of our manuscript. We amended the previous version of the manuscript in order to meet the received recommendations and suggestions. The changes made in the revised manuscript appear marked with Track Changes for an easy identification.

Please find below our detailed responses (point by point) to the comments:

Comment 1

I reviewed this work before, I advised it to be rejected. This decision was associated with the need to make a large number of changes in the presented manuscript.

The current version contains the expected changes, mostly regarding my previous comments. The text  is thoroughly reworded and supplemented with new data. Additional drawings have also been introduced. In this actual form, it seems to me that this work has become interesting and suitable for publication. Only relatively small patches remain to be done.

Autors should thoroughly review the text and remove errors such as, for example;

- used expressions such as; "PCL-blend-PEG blend films", should rather be "PCL-blend-PEG films" such errors appear, for example, in the description of  figure 7 and 8.

Answer 1

At Referee’s suggestion we thoroughly reviewed the whole text and removed all errors, especially the mentioned ones, from figures captions (7and 8).

Comment 2

- Figure 6,   in order to better understand the results of coating degradation process, autors shoul to establish the same scale and range in the relevant drawings (max mass loss 100%, max time 16 weeks), or place all functions (curves) on the one chart.

Answer 2 : According to the Referee’s requirements we modified figure 6 and its respective caption:

New caption of Figure 6 sounds like:

Figure 6. Relative mass loss curves illustrating the degradation behavior of polymeric coatings occurring during dynamic testing in SBF at 37°C. The corresponding degradation time is 16 weeks for: simple PCL; PCL-blend-PEG (3:1) and PCL-blend-PEG (1:3) (a) coatings, and 120 minutes for simple PEG coatings(b), respectively.

Comment 3

It seems to me also, that they should check whether the observed phenomenon of faster degradation of the coating made from the blend compared to the rate of degradation of the poly(e-caprolactone) coating is not associated with the practical total elution of PEG in the first phase of degradation, and then erosive changes arising on this way  was resulting in observed phenomenon of acceleration.

Answer 3

At Referee suggestion, we completed the Results Section (Section 3.4. Coatings degradation behavior; page 15; paragraph 1; 429-432 lines) by adding a new phrase that sounds as:

 „We emphasize that PEG total elution is the main mechanism occurring during the first stage of degradation, followed by the erosive changes of blended material, process mainly due to the hydrolytic alteration of PCL that occurs slowly over the entire evaluation period.

Reviewer 2 Report

The revised manuscript is suitable for publication,

Author Response

We would like to thank to the Referee for his consideration.